

# Effect of SARS-CoV-2 infection on host competing endogenous RNA and miRNA network

Selcen Ari Yuka and Alper Yilmaz

Department of Bioengineering, Yildiz Technical University, Istanbul, Turkey

## ABSTRACT

Competing endogenous RNAs (ceRNA) play a crucial role in cell functions. Computational methods that provide large-scale analysis of the interactions between miRNAs and their competitive targets can contribute to the understanding of ceRNA regulations and critical regulatory functions. Recent reports showed that viral RNAs can compete with host RNAs against host miRNAs. Regarding SARS-CoV-2 RNA, no comprehensive study had been reported about its competition with cellular ceRNAs. In this study, for the first time, we used the ceRNAnetsim package to assess ceRNA network effects per individual cell and competitive behavior of SARS-CoV-2 RNA in the infected cells using single-cell sequencing data. Our computations identified 195 genes and 29 miRNAs which vary in competitive behavior specifically in presence of SARS-CoV-2 RNA. We also investigated 18 genes that are affected by genes that lost perturbation ability in presence of SARS-CoV-2 RNA in the human miRNA:ceRNA network. These transcripts have associations with COVID-19-related symptoms as well as many dysfunctions such as metabolic diseases, carcinomas, heart failure. Our results showed that the effects of the SARS-CoV-2 genome on host ceRNA interactions and consequent dysfunctions can be explained by competition among various miRNA targets. Our perturbation ability perspective has the potential to reveal yet to be discovered SARS-CoV-2 induced effects invisible to conventional approaches.

## INTRODUCTION

More than 4.5 million people have died and millions were infected worldwide as of writing this manuscript due to COVID-19 caused by severe acute respiratory syndrome coronavirus 2, SARS-CoV-2 (*WHO, 2021*). Understanding the infection mechanism and causes of dysfunctions in infected cells in order to develop therapeutic strategies was of utmost importance in midst of the global pandemic. Basically, SARS-CoV-2 enters the cell through the binding of its structural protein S with the extracellular membrane proteins of the host cell (ACE2 and TMPRSS2). After the release of viral genomic RNA (ss+RNA) in the cell, the microenvironment is established for viral replication by expression of non-structural proteins and biogenesis of viral replication organelles (*V'kovski et al., 2020*; *de Wilde et al., 2017*). Following viral infection, regulations over the mechanisms that support the progression of the infection are undertaken (*V'kovski et al., 2020*;

Corresponding author
Selcen Ari Yuka,
selcenay@yildiz.edu.tr

*Alipoor et al., 2021*). As in most viral infections, SARS-CoV-2 RNA and proteins involve in interactions with host factors each of which should be considered in-depth to elucidate viral infection mechanisms.

One of the most critical interactions has a negative toll on host gene expression *via* miRNA:target RNA interactions. MiRNAs, the non-coding short RNAs, play important role in the regulation of cells through degradation or translational inhibition of the target RNA after transcription (*Bartel, 2004*; *Bartel, 2009*; *Brennecke et al., 2005*). Since each miRNA targets many mRNAs, miRNA:target interaction network emerges in which targets compete for targeting miRNA level. All targets competing for a single miRNA are called competing endogenous RNAs (ceRNA).

Since viral RNAs are targeted by host miRNAs or viral miRNAs can target host transcripts, viral RNAs should be incorporated in miRNA:target networks (*Serpeloni et al., 2021*; *Marchi et al., 2021*). More interestingly, there are studies showing that viruses sequester host miRNAs and interfere with their repression function. Such an interaction had been exhibited by the sequestration of human miR-122 by positive-sense single-stranded Hepatitis C Virus (HCV) (*Otto & Puglisi, 2004*). It had been reported that miR-122 interacts with the 5′end of viral RNA thus HCV RNA acts as a sponge for that miRNA and decreases its activity on other targets of the tumor suppressor miR-122 leading to the formation of hepatocellular carcinoma (*Shimakami et al., 2012*; *Luna et al., 2015*). Another case of viral RNA:host miRNA interaction had been reported in Pestiviruses where miR-17 sequestration by viral RNA was shown to de-repress cellular miR-17 targets (*Scheel et al., 2016*). Cases of the more complex interplay between viral RNA and host miRNAs are reviewed in *Gebert & MacRae (2018)*. These findings suggest that subsequent effects of host miRNA sequestration by viral RNAs should be studied in detail.

Using various prediction algorithms, multiple studies had reported that SARS-CoV-2 genomic RNA is targeted by human miRNAs (*Nersisyan et al., 2020*; *Khan et al., 2020*; *Chow & Salmena, 2020*; *Demirci & Adan, 2020*). Although such studies omitted analysis of targeting miRNAs in the context of ceRNA network, two recent studies addressed the effect of the viral genome on ceRNA interactions. *Arora et al. (2020)* studied the ceRNA network based on differentially expressed genes (DEGs) on the SARS-CoV infected mouse mRNA dataset. Based on DEGs, researchers addressed miRNA:mRNA, miRNA:circRNA, miRNA:lncRNA, and mRNA/transcription factor interactions and tried to identify critical ncRNA-mRNA-TF axes. However, that study ignored the sponge effect of viral RNA against the human ceRNA network. Proper ceRNA analysis for SARS-CoV-2 infection, decreasing miRNA activity caused by viral RNAs and perturbation in ceRNA network, was proposed and initially attempted by *Arancio (2020)*.

Studying the interplay between mRNAs and miRNAs is challenging since a single miRNA targets multiple mRNAs and on the other hand, a single mRNA is targeted by several miRNAs. In this study, our objective is determining the effect of SARS-CoV-2 viral RNA within human large-scale miRNA:target interactions using our previously developed network-based tool (*Yuka & Yilmaz, 2020*; *Yuka & Yilmaz, 2021*). Briefly, the method is based on determining the repression efficiency of miRNAs on their targets proportional to their expression. Change in the expression level of a target affects the repression activity

of miRNAs on other targets due to competition. Moreover, due to the existence of shared targets between multiple miRNAs in a large-scale network, the effect of competition cascades through the whole ceRNA network.

By utilizing our tool, we studied the competition of SARS-CoV-2 RNA with human ceRNAs and the consequences of that competition. Single-cell RNA sequencing and total small RNA sequencing datasets from SARS-CoV-1 and SARS-CoV-2 infected lung cancer cells (Calu3) were used to construct the ceRNA network in infected cells. MiRNAs potentially targeting the SARS-CoV-1 and/or SARS-CoV-2 genome were integrated into the ceRNA network. Afterward, we performed simulations to reveal human gene transcripts which were affected by competition in presence of viral RNA. We identified genes de-repressed by miRNAs targeting SARS-CoV-2 RNA. Our tool has the potential to include other players in the ceRNA network (lncRNA, circRNA, etc.) provided that high-throughput sequencing data is available in the future.

## MATERIALS AND METHODS

### Data
Single-cell RNA-Seq and bulk miRNA-Seq data of Severe acute respiratory syndrome-related coronavirus (SARS-CoV-1) and Severe acute respiratory syndrome coronavirus (SARS-CoV-2) infected human cell lines were downloaded from Single Cell Expression Atlas (accession number PRJNA625518) (*Wyler et al., 2021*; *Papatheodorou et al., 2019*) and Gene Expression Omnibus (GEO accession no GSE148729) (*Barrett et al., 2012*), respectively. miRNA-Seq reading data had been normalized to reads per million. To construct miRNA:target interaction network, strongly supported miRNA and gene target pairs were downloaded from miRTarBase (version 8) (*Huang et al., 2019*) which houses experimental miRNA:target pair data. Human miRNAs targeting SARS-CoV-2 genome were extracted from four previous reports which utilized various prediction algorithms (*Khan et al., 2020*; *Chow & Salmena, 2020*; *Demirci & Adan, 2020*; *Fulzele et al., 2020*). The miRNAs common in at least three studies were considered as targeting the SARS-CoV-2 genome. SARS-CoV-1 targeting miRNAs were extracted from the study of *Khan et al. (2020)*.

As viral RNA read counts, we considered the number of mapped reads (per million reads) of viral RNAs (*i.e.*, SARS-CoV-1/2) at 12 h post-infection in Calu3 cells (*Wyler et al., 2021*) to simulate its competitive behavior with human gene transcripts. Viral RNA read counts were normalized to viral genome length in kb (*i.e.*, RPKM). The single-cell RNA-Seq data and bulk miRNA-Seq data from SARS-CoV-1/2 infected Calu3 cells were combined according to miRNA:target pair dataset from miRTarBase (*Huang et al., 2019*). The perturbation simulations were carried out with and without viral RNAs.

### ceRNA network simulations in single-cell datasets
The ceRNA network simulations were carried out using the ceRNAnetsim R package we had developed previously (*Yuka & Yilmaz, 2021*; *Yuka & Yilmaz, 2020*). Briefly, each node in the network was used as a trigger for perturbation after which perturbation ability in the whole network was calculated. Node importance was evaluated by calculating the average

change in expression for all nodes in miRNA:target network when that node was perturbed. Simulation with 2,800 cells for each group (*i.e.,* SARS-CoV-1, SARS-CoV-2, and Mock samples) that were sampled, was performed with 20 iterations after 3 fold up-regulation of each node in parallel (*Tange, 2011*). SARS-CoV-1/2 dataset simulations were performed in miRNA:target network with/without viral RNAs. Viral RNA was incorporated into the ceRNA network through miRNAs targeting the viral RNA.

### Detecting miRNA:target interactions disturbed in presence of viral RNAs

We performed perturbation simulation using expression data for each cell in the ceRNA network in the absence and presence of viral RNA and calculated the perturbation ability of each node. Threshold value to be considered perturbing node was calculated by finite mixture model in results of all cell types (mock, SARS-CoV-1 infected, SARS-CoV-2 infected) (*Trang et al., 2015*). By comparing the perturbation ability of nodes in the presence and absence of SARS-CoV-1/2 RNA, we revealed nodes with altered perturbation ability in presence of SARS-CoV-1 or SARS-CoV-2.

In order to find nodes that are affected by genes with altered perturbation ability, we performed simulations for each gene with cells in which the gene lost its perturbation. Simulations of genes that lost perturbation ability in SARS-CoV-2 infection were carried out under the same conditions (20 iterations after 3 fold up-regulation) to find the genes affected by perturbation of these nodes. Functional annotation analysis was performed with DAVID and TAM 2.0 functional annotation tools were for genes and miRNAs, respectively (*Jiao et al., 2012*; *Li et al., 2018*).

## RESULTS

### Single-cell expression data overlaid on ceRNA network originated from miRTarBase

Single-cell RNA (sc-RNA) sequencing data of Calu3 cells at 12 h after infection contained 25,757 gene transcripts for 3,779 SARS-CoV-1 infected cells and 14,283 SARS-CoV-2 infected cells. In addition, sc-RNA sequencing data contained 10,501 mock cells at 12 h. Bulk reading counts for 1,541 miRNAs were available at 12 h from small RNA-Seq data. After integrating all the data, a ceRNA network comprised of 9,009 miRNA:target interactions among 739 miRNAs and 3,032 targets were constructed (Table 1). For each single cell, the expression data was overlaid on the ceRNA network prior to simulations.

For ceRNA networks containing SARS-CoV RNA, existing networks were extended with miRNAs potentially target viral RNAs (*Khan et al., 2020*; *Chow & Salmena, 2020*; *Demirci & Adan, 2020*; *Fulzele et al., 2020*). 129 miRNAs targeted SARS-CoV-1 RNA, 39 of which have already been in the ceRNA network, so the network was extended by 90 miRNAs. For SARS-CoV-2, 51 miRNAs targeted SARS-CoV-1 RNA, 14 of which have already been in the ceRNA network, so the network was extended by 37 miRNAs (Fig. S1). The contents of extended ceRNA networks are summarized in Table 1.

Since mock cells lacked miRNA reads at 12 h, we performed differential expression analysis using DEseq2 package with miRNA reads (TPM) at 4 and 24 h (*Love, Huber &*
**Table 1** The summary of ceRNA networks constructed for simulations.

|  | Mock | w/ SARS-CoV-1 | w/ SARS-CoV-2 |
| --- | --- | --- | --- |
| No of Interactions | 9,009 | 9,138 | 9,060 |
| miRNA | 739 | 829 | 776 |
| Target | 3,032 | 3,033 | 3,033 |

*Anders, 2014*). We found that 13 (miR-135b-5p, miR-141-3p, miR-186-5p, miR-18a-5p, miR-18b-5p, miR-19a-3p, miR-20a-5p, miR-29a-3p, miR-29c-3p, miR-30e−5p, miR-424-5p, miR-539-3p, miR-582-5p) of 1541 miRNAs were differentially expressed between 4 and 24 h in Mock cells (Fig. S2). Since the number of differentially expressed miRNAs was negligible, we considered 24-hour mock count data as comparable with 12-hour infection data.

## Perturbation ability of miRNAs and genes in calu3 cells changes upon infection

Simulations in 2,800 of each type of cells (*i.e.,* mock, CoV-1- and CoV-2-infected) revealed that particular miRNA and gene nodes in ceRNA networks had altered perturbation abilities in the absence and presence of viral RNA. Since the simulations were performed in many cells, the finite mixture model was used to determine the minimum number of cells required for a gene or miRNA node to be considered having perturbation ability. Since the perturbation simulations were performed in many cells, a cut-off, the minimum number of cells, was calculated above which the node is considered significantly perturbing. Perturbing miRNA and target numbers indicate the number of miRNAs and target genes which are above the corresponding cut-off value (Table 2).

We performed ceRNA network simulations of Mock, and infected cells with and without viral RNAs (CoV-1 and CoV-2). When we compared perturbation simulation results of three ceRNA networks (Mock, Infected without viral RNA, Infected with viral RNA) for each SARS-CoV, pairwise Pearson correlations of Mock *vs* w/o SARS-CoV-1 RNA and Mock *vs* w/o SARS-CoV-2 were calculated as 0.96 and 0.91, respectively (Figs. S3–S4). Thus, analysis of changed perturbation ability was performed on results from networks with and without viral RNA only.

As a result of perturbation simulations in the ceRNA network of SARS-CoV-1 infected cells, it was observed that SARS-CoV-1 RNA was perturbing in almost all cells (2779 of 2800), while 6 genes (HMBOX1, MYCN, IGFBP3, DIRAS3, IKZF1, ZNF763) lost their perturbation ability in presence of SARS-CoV-1 RNA in ceRNA network (Supplementary Data). Interestingly, 7 miRNAs (miR-1229-3p, miR-1307-3p, miR-182-5p, miR-22-5p, miR-30c-1-3p, miR-425-3p, miR-550a-5p) gained perturbation ability after SARS-CoV-1 integration (Table 3).

On the other hand, trends of changes in perturbation ability in SARS-CoV-2 infected cells were drastically different. It was observed that 195 genes lost their perturbation ability in the SARS-CoV-2 RNA integrated ceRNA network simulations (Supplementary Data). In contrast to SARS-CoV-1 results, 29 miRNAs gained perturbation ability in presence of SARS-CoV-2 RNA (Table 3). Please note that no miRNA with lost perturbation ability was
**Table 2  The cut-off for cell numbers and perturbing node counts above the cut-off for each cell type according to the finite mixture model.**

|  | Mock | SARS-CoV-1 | | SARS-CoV-2 | |
|---|---|---|---|---|---|
|  |  | w/o CoV-1 | w/ CoV-1 | w/o CoV-2 | w/ CoV-2 |
| miRNA cut-off (cells) | 261 | 309 | 293 | 156 | 80 |
| Perturbing miRNA | 122 | 142 | 150 | 145 | 174 |
| Target cut-off (cells) | 68 | 52 | 53 | 42 | 110 |
| Perturbing Target | 938 | 1070 | 1065 | 1039 | 845 |

**Table 3  MiRNAs gained perturbation abilities in the presence of SARS-CoV-1 and SARS-CoV-2 RNAs.**

| SARS-CoV-1 | SARS-CoV-2 |
|---|---|
| miR-425-3p, miR-182-5p, miR-1229-3p, miR-1307-3p, miR-550a-5p, **miR-30c-1-3p**, **miR-22-5p** | let-7e−3p, miR-1260b, miR-1270, miR-140-3p, miR-146a-5p, miR-15a-3p, miR-188-5p, miR-194-3p, miR-22-3p, miR-23b-5p, miR-29a-5p, miR-379-5p, miR-409-3p, miR-424-3p, miR-501-5p, miR-539-5p, miR-548b-3p, miR-551a, miR-589-5p, miR-628-5p, miR-629-5p, miR-642a-5p, miR-652-3p, miR-660-5p, miR-766-3p, miR-873-5p, miR-935, **miR-22-5p**, **miR-30c-1-3p** |

detected in simulations in presence of both viral RNAs. Only one gene and 2 miRNAs were found to be common in the list of nodes that had altered perturbation ability in presence of each SARS-CoV RNAs. That gene and two miRNAs were listed in Tables S1–S2.

## Functional analysis of genes losing perturbation ability only in SARS-CoV-2 infection

A total of 195 genes that lost perturbation ability specifically in presence of SARS-CoV-2 RNA were subject to functional analysis. The genes which lost perturbation ability showed enrichment in several important KEGG pathways such as p53 signaling pathway (hsa04115, 8.48 fold enrichment with 0.000002 $p$-value), Hepatitis B (hsa05161, 4.31 fold enrichment with 0.0002 $p$-value), and Cytokine-cytokine receptor interaction (hsa04060, 3.27 fold enrichment with 0.0003 $p$-value). We observed that they enriched in many molecular functions as transmembrane receptor protein serine/threonine kinase activity (GO:0004675, 41.91 fold enrichment with 0.0001 $p$-value), protein kinase activity (GO:0004672, 3.415026 fold enrichment with 0.0004 $p$-value), and activin binding (GO:0048185, 39.294693 fold enrichment with 0.000005 $p$-value). Also, in important biological processes such as negative regulation of ERK1 and ERK2 cascade (GO:0070373, 12.38 fold enrichment with 0.000003 $p$-value), positive regulation of protein kinase B signaling (GO:0051897, 8.55 fold enrichment with 0.00004 $p$-value), and response to drug (GO:0042493, 4.73 fold enrichment with 0.000001 $p$-value) significant enrichment were determined. All data is available in the Supplementary Data.

In order to decipher the potential downstream effects of SARS-CoV-2 RNA in the ceRNA network, we identified genes that are affected through the ceRNA network by 195 genes (Fig. 1A). First, we collected barcodes of cells where perturbation was lost for each of the 195 genes in the presence of SARS-CoV-2 RNA. Then we performed simulations triggered

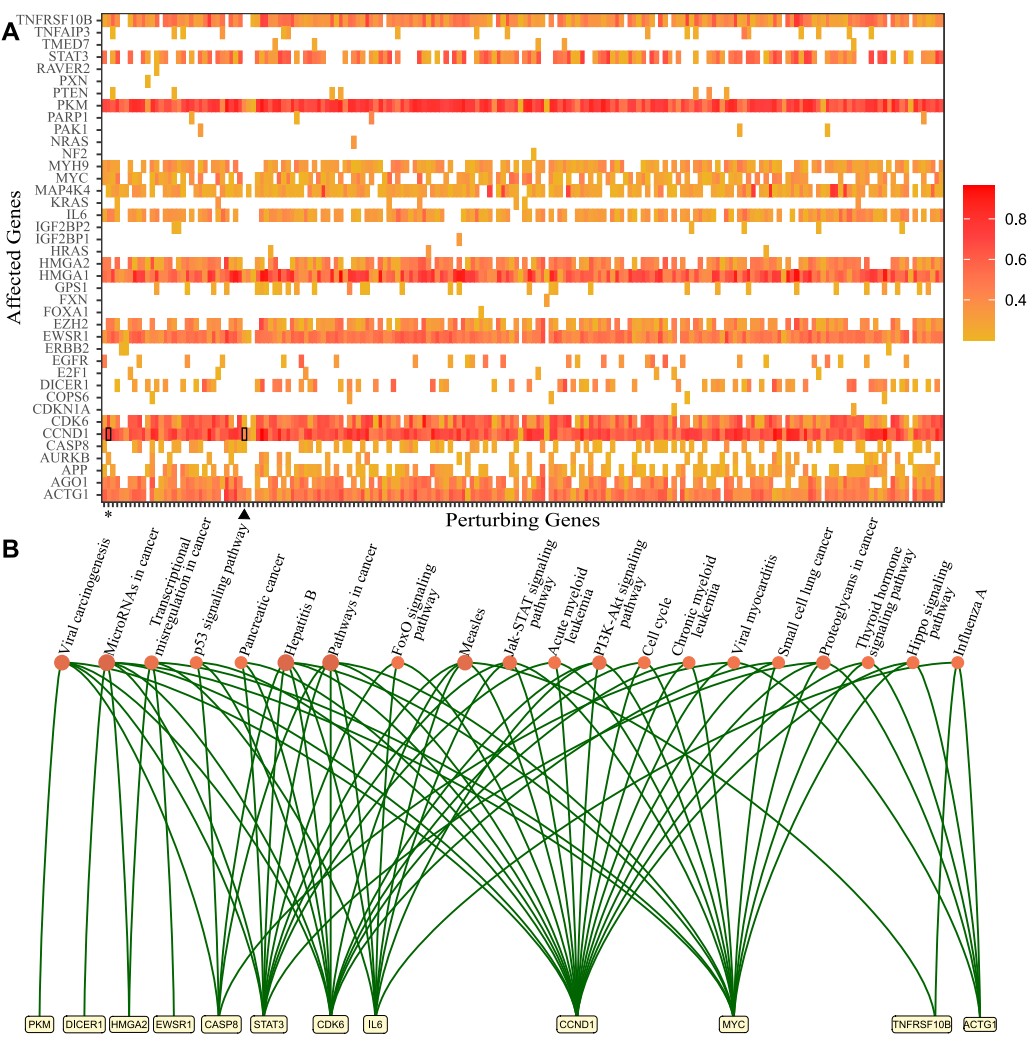

**Figure 1** **Genes that lost perturbation ability in presence of viral RNA and genes affected by those genes through the ceRNA network.** (A) The genes which lost perturbation ability in the presence of SARS-CoV-2 RNA in the ceRNA network were presented in the *x*-axis (Perturbing Genes). The genes affected *via* ceRNA network competitions by perturbing genes are presented in the *y*-axis. The heatmap summarizes ceRNA network simulations in many cells for each perturbing gene and the ratio of perturbed cases in simulated cells. Heatmap results for ACVR1B (asterisk) and CD47 (triangle) were highlighted. (B) The significantly enriched KEGG pathways and genes which were affected commonly by 195 genes that lost perturbation in presence of SARS-CoV-2 (*p*-value <0.05).

from each gene and found the most affected genes from perturbation simulations. For example, in simulations using 102 cells for the ASVR1B gene, CCND1 gene was affected in 91 cells (asterisk, ratio 0.89). In the simulations performed with the CD47 gene in 103 cells, we found that CCND1 gene was among the most affected genes in 56 cells (triangle, ratio 0.54) (Fig. 1A). Ultimately, 18 genes (TNFRSF10B, STAT3, PKM, MYH9, MYC, MAP4K4, IL6, HMGA1, HMGA2, E2H2, EWSR1, DICER1, CDK6, CCND1, CASP8, APP, AGO1, ACTG1) appear to be affected by almost all 195 genes in absence of SARS-CoV-2 RNA and were no longer perturbed in presence of SARS-CoV-2 RNA.

Functional annotation analysis of the affected genes *via* DAVID (*Jiao et al., 2012*) showed enrichment in several important KEGG pathways (Fig. 1B). For instance, Hepatitis B (hsa05161, 18.98 fold enrichment with 0.000007 *p*-value), measles (hsa05162, 17.24 fold enrichment with 0.0001 *p*-value), microRNAs in cancer (hsa05206, 9.62 fold enrichment with 0.0001 *p*-value), and viral carcinogenesis (hsa05203, 11.18 fold enrichment with 0.0006 *p*-value) were among pathways with highest enrichment. Interestingly, these genes were significantly enriched in biological processes such as response to estradiol (GO:0032355, 43.42 fold enrichment 0.00008 *p*-value) and oncogene-induced cell senescence (GO:0090402, 987.76 fold enrichment with 0.001 *p*-value). All of the genes were also enriched significantly in molecular functions such as protein binding (GO:0005515, 1.92 fold enrichment with 0.00002 *p*-value) and transcription factor binding(GO:0008134, 17.48 fold enrichment with 0.0001 *p*-value). Full functional annotation data of affected genes is available in the Supplementary Data.

Our analysis revealed that specifically in presence of SARS-CoV-2 RNA, several genes lose their perturbation ability in the ceRNA network, rendering particular genes uncontrollable by the ceRNA network which were shown to be associated with diseases or cellular functions in literature. We found three genes (CASP8, ACTG1, and CCND1) that were significantly associated with viral myocarditis. Although several reports highlighted the issue of SARS-CoV-2-induced viral myocarditis, evidence for the involvement of these genes in the onset of viral myocarditis is lacking (*Siripanthong et al., 2020*). Although its molecular mechanisms have not been explained yet, studies on COVID-19 patient samples had reported the loss of thyroid function (CCND1, MYC, ACTG1 genes in Thyroid hormone signaling pathway with 11.96 fold enrichment and 0.02 *p*-value, was observed) and related loss of immune system regulation (*Lui et al., 2020*; *Khoo et al., 2021*). SARS-CoV-2-induced viral carcinogenesis has not been addressed thoroughly yet but in a recent study, the effect of the virus on tumor development had been investigated using SARS-CoV-2-host networks and the potential role of virus for tumorigenesis had been suggested (*Souchelnytskyi, Nera & Souchelnytskyi, 2021*). Interestingly, we found that 4 genes (CASP8, CCND1, MYC, and STAT3) were enriched in the biological process of response to estradiol. In the literature, it had been reported that estradiol may be a therapeutic option in SARS-CoV-2 viral infection due to the immune regulatory function (*Mauvais-Jarvis, Klein & Levin, 2020*). For detailed functional annotation and enrichment results please refer to the Supplementary Data. Although there are many recent studies on the systemic effects or symptoms of SARS-CoV-2 infection, number of studies that dealt with SARS-CoV-2-induced regulatory mechanisms on a large scale network is quite limited.

## Functional annotations of miRNAs that gained perturbation ability in the presence of SARS-CoV-2 RNA

We observed that 29 miRNAs gained perturbation ability as a result of the integration of SARS-CoV-2 RNA to the host ceRNA network. By using TAM2.0 database (*Li et al., 2018*), we observed that miRNAs were significantly enriched in a wide variety of diseases such as
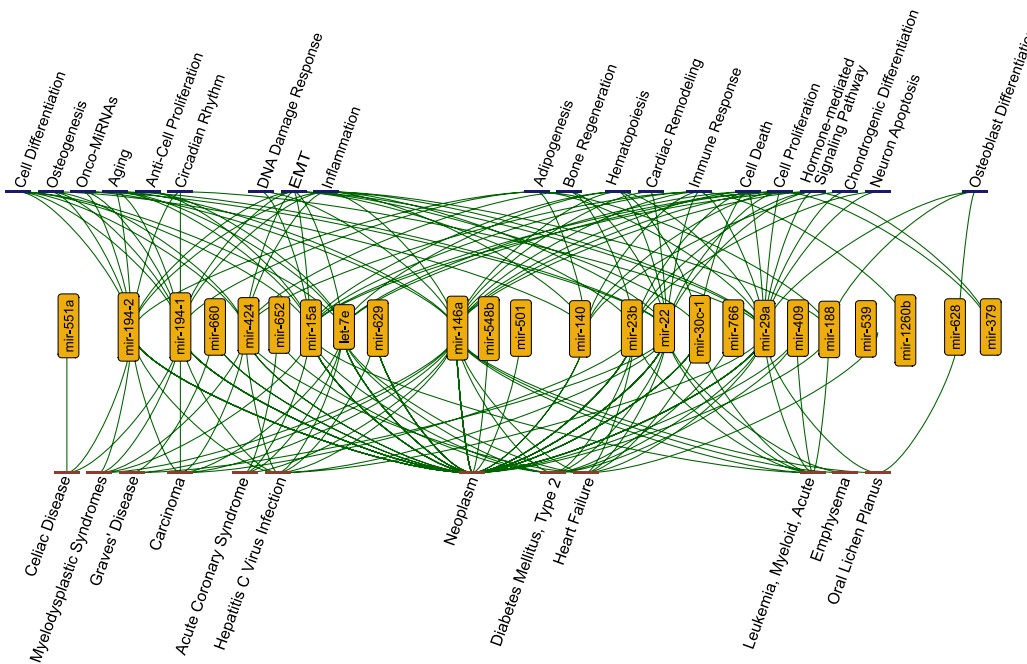

**Figure 2** **The functions (blue nodes, top) and diseases (red nodes, bottom) having enriched miRNAs (yellow nodes) that gained perturbation ability in presence of SARS-CoV-2 RNA.** The network shows annotations from TAM2.0 database. All data is available in the Supplementary Data.

metabolic disorders and carcinoma. They were also enriched in many cellular functions such as inflammation and immune response (Fig. 2).

Strikingly, we observed that several functions and diseases associated with miRNAs overlapped with symptoms of COVID-19 cases. For example, miR-23b, miR-30c-1-3p, miR-424-3p, miR-22, miR-146a, let-7e, miR-539, and miR-29a were found to be associated with heart failure (4.08 fold enrichment with 0.0004 *p*-value) which accompanies SARS-CoV-2 infections (*Tomasoni et al., 2020*). miR-22 and miR-146a were significantly enriched in emphysema, shortness of breath, (61.28 fold enrichment with 0.0002 *p*-value) which was reported as a strong complication of SARS-CoV-2 infection (*Lacroix et al., 2020*). Previous reports had shown that miR-22 controls the activation of immune system cells through the activation of histone deacetylase HDAC4, while miR-146a promotes the generation of abnormal inflammatory responses through a fibroblast-mediated way (*Lu et al., 2015*; *Sato et al., 2010*). These miRNAs may contribute to the development of chronic inflammatory disease through different mechanisms. It was reported that SARS-CoV-2 infection had the potential to cause Diabetes Mellitus, Type 2 through damage to $\beta$-cells (*Hayden, 2020*) and several miRNAs (miR-766, miR-652, miR-29a, miR-409, miR-23b, miR-15a, miR-22, miR-146a, let-7e) were significantly enriched in this disease. Our results pointed out direct or indirect associations between COVID-19 related diseases and the host ceRNA network.

Although there is not full knowledge about the molecular mechanisms regarding destructive effects of COVID-19, miRNA interactions of viral infection may have significant
contributions. Considering the miRNA:ceRNA interactions in SARS-CoV-2 infections may facilitate the understanding of the systemic dysfunctions caused by viral RNA.

## DISCUSSION

With the discovery that miRNA and target interactions are important in many cellular functions, studies of the competitive behavior of miRNA targets had started to attract attention. However, calculating the network-wide effects of a single node in a large network with many-to-many interactions (i.e, between miRNAs and targets) is proved to be a daunting challenge. On the other hand, several studies had reported that viral genomes play a crucial role in the disruption of miRNA:target interactions in the host cell and the mechanisms of dysfunction or viral-induced diseases observed after viral infections can be explained (*Shimakami et al., 2012*; *Scheel et al., 2016*). SARS-CoV-2 causes many dysfunctions which are needed to be explained. However, multifaceted large-scale analyzes are needed to understand the mechanisms of the systemic effects of SARS-CoV-2 infection. Since SARS-CoV-2 RNA is targeted by host miRNAs, viral RNA participates as ceRNA in host miRNA:ceRNA interactions. Therefore, we developed a network-based model that handles complex interactions and applied our approach to find the regulations that can be disrupted by SARS-CoV-2. Our results showed that SARS-CoV-2 viral RNA has the potency to perturb host miRNA:ceRNA interaction and in presence of viral RNA, particular host genes lose perturbation ability.

Since bulk RNA-Seq data contains accumulated reads from cells that masks individual gene expression landscape, we utilized 2800 single-cell RNA sequencing data for each mock, SARS-CoV-1, and SARS-CoV-2 infected samples. A total of 195 genes lost perturbation ability and 29 miRNAs gained perturbation ability in the presence of SARS-CoV-2 RNA. As a result of our simulations with these 195 genes, 18 genes were perturbed in many cells and were subject to functional annotation analysis (Fig. 1). Twenty-eight miRNAs enriched in different diseases ranging from immune diseases to metabolic disorders. Interestingly, when the genes and miRNAs that altered perturbation in presence of SARS-CoV-2 RNA were plotted as a network, the nodes had no direct connection to virus RNA (Fig. 3), suggesting that ceRNA networks harbor distant effects. In earlier studies, it was shown that viral RNA (eg. HCV, pestiviruses) sequestered a host miRNA causing de-repression of its targets (*Shimakami et al., 2012*). These approaches are limited by direct binding events exhibiting a near-sighted perspective. Our approach considers the whole ceRNA network and reveals indirect interplay in a comprehensive manner.

Our package has the additional advantage of integrating various players into calculation in the large-scale ceRNA network. Although *Demirci & Adan (2020)* predicted viral miRNAs, generated by SARS-CoV-2, we were not able to integrate them during calculations since high-throughput expression data was not available for such miRNAs. Our approach has the ability to integrate ceRNA players such as lncRNAs and circRNAs which are found in miRNA:ceRNA interactions (*Salmena et al., 2011*). However, due to the lack of single-cell sequencing data for such ceRNA players in SARS-CoV-2 infected cells, we could only consider mRNA and miRNA levels in our calculations. When high-throughput sequencing

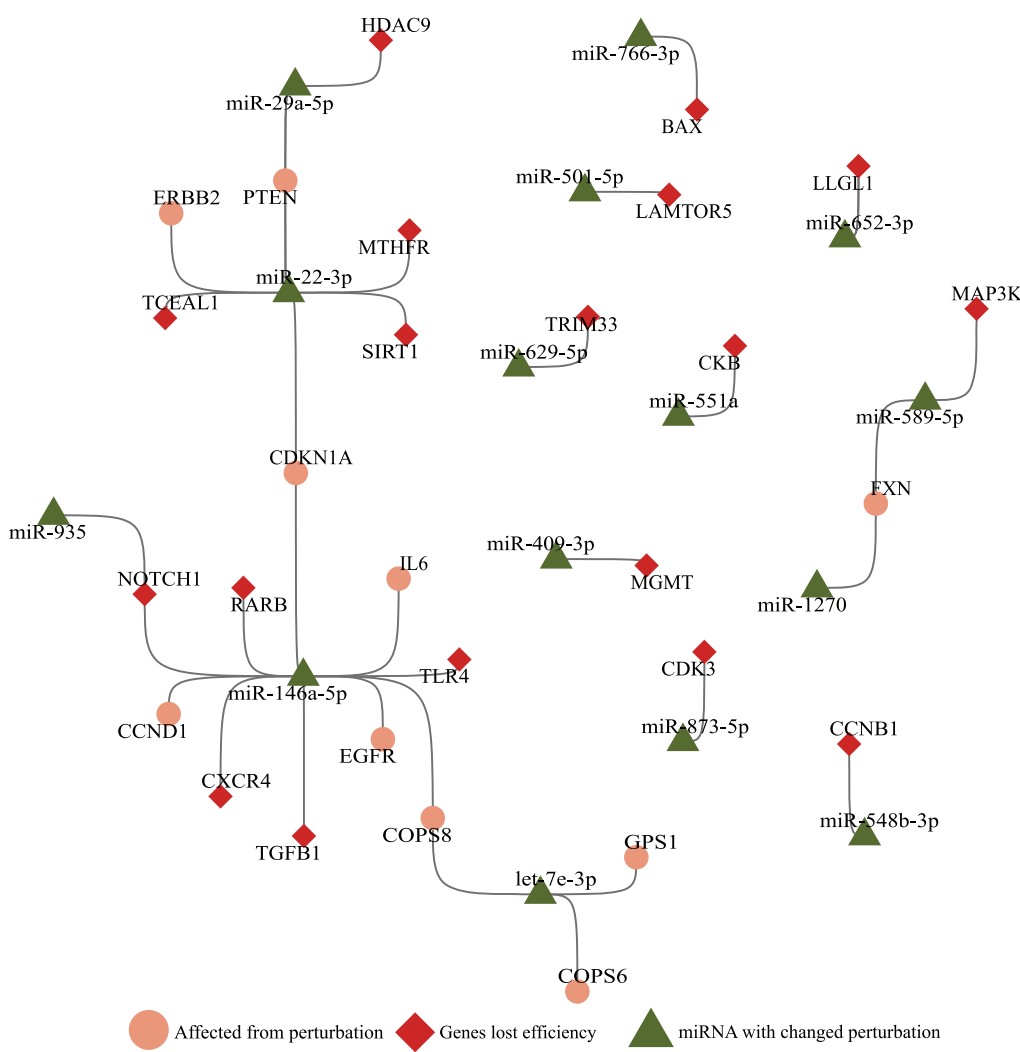

**Figure 3   Illustration of SARS-CoV-2 RNA effects on ceRNA network.** miRNAs (green triangle), genes (red diamond) that altered perturbation ability were extracted, and genes (salmon circle) affected by genes with altered perturbation were added to construct a network. Nodes without any edges (isolated nodes) were not shown for clarity. Genes affected in the small number of cells (Ratio < 0.1, Fig. 1A) were not included.

data for all transcripts encompassing the non-coding RNA world becomes available, our package would perform more accurate miRNA:ceRNA network calculations.

In order to integrate SARS-CoV-2 RNA into the ceRNA network, we had to collect host miRNAs that target viral RNA. There are numerous studies predicting such miRNAs (*Khan et al., 2020*; *Chow & Salmena, 2020*; *Demirci & Adan, 2020*; *Fulzele et al., 2020*). However, we faced an unpleasant situation where the consensus among all studies is almost absent (Fig. S1). Such inconsistency in predictions suggests that high-throughput experimental validation methods (CLASH, CLEAR-CLiP) are needed to discover the functional interactions of host miRNAs with the viral target. Due to lack of consensus among prediction datasets, we constructed the ceRNA network using predicted miRNAs
that are common in at least three out of four studies (*Khan et al., 2020*; *Chow & Salmena, 2020*; *Demirci & Adan, 2020*; *Fulzele et al., 2020*).

With the integration of SARS-CoV-2 RNA into the host miRNA:ceRNA network, results of our simulations were consistent with the literature and have the potential to reveal new perspectives on the molecular mechanisms of the infection. Most of SARS-CoV-2-induced dysfunctions may be due to inflammation from cytokine storm and immunological discordance (*Tao et al., 2020*). It had been reported that cytokine storm is triggered through activation of NF-$\kappa$B *via* pro-inflammatory genes (IL-6/STATs) activated in the presence of viral infection (*Mahmudpour et al., 2020*; *Murakami & Hirano, 2012*). Consistent with previous studies, STAT3 and IL-6 were detected in our analyzes among the genes most affected by the presence of SARS-CoV-2 (Fig. 1A). It had been reported that miR-146a targets IL-6 and was downregulated in COVID patients (*Sabbatinelli et al., 2021*). Our results suggests alternative scenarios of IL-6 regulation in presence of viral RNA. For example, the miR-935/NOTCH1/miR-146a/IL-6 or TGFB1/miR-146a/IL-6 axes may also have indirect effects on IL-6 levels (Fig. 3).

We observed enrichment of many cancer-related functions and pathways in both our miRNA and gene functional annotations which is consistent with earlier studies associating COVID-19 and tumorigenesis. For instance, expression of oncogenes (MYC, ERBB2, CCND1) that are used as diagnostic and prognostic markers and genes important in cell proliferation (TGF $\beta$) were affected by SARS-CoV-2 infection suggesting that viral infection is critical to tumorigenesis (*Souchelnytskyi, Nera & Souchelnytskyi, 2021*). Moreover, it had been reported that either viral genomes may interfere with host miRNAs or viral miRNAs may regulate host genes and play a role in carcinoma development (*Gebert & MacRae, 2018*; *Siniscalchi et al., 2021*; *Gallo et al., 2020*).

There are studies in which miRNAs are suggested as therapeutic components due to their interactions with the viral genome (*Ying et al., 2021*; *Zhang et al., 2021*). For example, anti-HIV potential for miR-29a has been reported due to its interaction with HIV Nef viral protein and viral genome (*Nathans et al., 2009*; *Ahluwalia et al., 2008*). For such approach, merely finding a miRNA targeting the viral genome wouldn't suffice due to complex ceRNA interactions. In order to find human miRNAs targeting the SARS-CoV-2 genome for therapeutic purposes, many-to-many miRNA:target interactions in infected cells should be considered. Our method is a novel approach to predict the aftermath of either miRNA or ceRNA level changes on the ceRNA network. For example, miR-29a, which has been suggested as a therapeutic option for HIV, is one of the miRNAs affected by the presence of SARS-CoV-2, although it interacts with the tumor suppressor PTEN in our analysis. The effect of the therapeutic use of miR-29a on a metabolic regulator and tumor suppressor gene needs to be considered (*Chen et al., 2018*). So, finding orthogonal therapeutic options is crucial so that the normal functions of the cell are not affected abnormally.

The role of miRNA-mediated strategies or role of interactions between viral RNA and host miRNAs during SARS-CoV-2 infection is unexplored. However, there is only one network-based study aimed at the analysis of ceRNA network encompassing multiple players (i.e, circRNA, lncRNA). *Arora et al. (2020)* identified differentially expressed genes in SARS-CoV infected mouse cells and then integrated miRNAs targeting those genes

and additional ceRNAs without any expression data. However, such topological approach which ignores the competition between various RNA species is insufficient to perform realistic ceRNA calculations.

There are many recent studies evaluating the symptoms and systemic effects of SARS-CoV-2. However, studies integrating omics data in a network-based approach usually perform computations on nodes and their immediate neighbors thus overlooking network-wide effects of individual nodes. In contrast, in ceRNA networks, each node has the potential to affect the whole network. Our R package, based on ceRNA network interactions, aims to predict network-wide effects of expression level change in a single node. Also, our tool is capable of integrating parameters that affect affinity or degradation efficiency of miRNA, thus when such data becomes available in the future for SARS-CoV-2 and human miRNAs, our tool can incorporate for more accurate calculations.

## CONCLUSIONS

Our results highlight the effects of SARS-CoV-2 RNA on the human miRNA:target interaction network. It offers a new perspective for understanding the molecular mechanisms of systemic effects of SARS-CoV-2 infection, and the competitive behavior between human miRNA targets and the viral RNA. Canonical approaches for gene expression analysis involve differentially expressed gene (DEG) analysis to which network-wide and distant ceRNA network interactions are invisible. The perturbation ability perspective suggests that a gene or miRNA can gain or lose perturbation ability without changing its expression level due to complex ceRNA network cross-talk. Thus, changes in perturbation ability in ceRNA networks should be considered for more accurate interpretation. Our results have the potential to form a basis for possible future therapeutic approaches to overcome virus-induced systemic disorders.

## ACKNOWLEDGEMENTS

The numerical calculations reported in this paper were partially performed at TUBITAK ULAKBIM, High Performance and Grid Computing Center (TRUBA resources).

### Funding

The authors received no funding for this work.

### Competing Interests

The authors declare there are no competing interests.

### Author Contributions

- Selcen Ari Yuka conceived and designed the experiments, performed the experiments, analyzed the data, prepared figures and/or tables, authored or reviewed drafts of the paper, and approved the final draft.

- Alper Yilmaz conceived and designed the experiments, analyzed the data, authored or reviewed drafts of the paper, and approved the final draft.

## Data Availability

The significant genes and miRNAs from simulation results are available in the Supplementary Files.

## Supplemental Information

Supplemental information for this article can be found online at http://dx.doi.org/10.7717/peerj.12370#supplemental-information.

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
