# Peer review of "Effect of SARS-CoV-2 infection on host competing endogenous RNA and miRNA network"

_PeerJ, doi:10.7717/peerj.12370_

## Round 0.1 · original submission · Major Revisions

The manuscript investigates an important topic but the scientific writing can be improved. Provide a stronger rationale why is this research needed and the significance of the research. The title of table 2 is very large, try to use a shorter title, mention the limitations of the research in the discussion part.

Reviewer 1 ·

Basic reporting

The authors have reported an important finding on the competing endogenous RNAs (ceRNA). Overall, the manuscript is well written. However, I have some comments:

Abstract: It is well written and structured. But, please mention the gaps in the previous study in line and the strengths of the new finding briefly.

Introduction: It is well written but very briefly. The authors should focus on details of the biology, such as what are mRNAs and miRNAs? Classify RNAs and their relations to SARS-CoV2 viral replication. Please state the previous study limitations and how this one will overcome those, and clearly write the objective of the study. You should not cite other papers in the last paragraph of any introduction. The authors have missed a couple of recent vital pieces of literature, such as this very interesting and high-impact one: https://doi.org/10.1016/j.gene.2020.145057.

DIscussion: Please add a brief hypothesis and findings in the first paragraph. Mention the strengths and limitations of the study.

Experimental design

No comments.

Validity of the findings

The findings are valid and logical to the writing flow.

Reviewer 2 ·

Basic reporting

The manuscript aims to investigate effects of SARS-CoV-2 infection on host competing endogeneous RNA and miRNA network through application of their previously developed R package.
Although it is a well written manuscript, there are various points needed to be improved:

1. According to Salmena et. al. 2011 paper, the ceRNA hypothesis which is also mentioned in the title of the submission, “a unifying hypothesis about how messenger RNAs, transcribed pseudogenes, and long noncoding RNAs “talk” to each other using microRNA response elements (MREs) as letters of a new language”. In addition to this, circular RNAs also have miRNA binding sites. This study should provide information/results about the other players of ceRNA rather than looking only at host mRNA and viral RNA targets of host miRNAs.

2. In the introduction section current knowledge about miRNAs in SARS-CoV-2 infection are explained in detail with many citations. The results of those cited papers should be compared and discussed with original findings of the study.

Experimental design

3. A major concern about the experimental design is the authors do not perform any predictions about miRNAs and their targets. It is understandable to obtain experimentally validated miRNA – mRNA targets from mirtarbase but human miRNAs targeting of SARS-CoV-1 and/or SARS-CoV-2 were extracted from a previous study. If existing datasets would be used, data from more sources should be considered. There are various publications for SARS-CoV-2 and human miRNAs, some are also available in the references list.

4. In line with the 3rd point, there are reports of SARS-CoV-2 encoded miRNAs, their viral and human targets. These should also be included in a ceRNA based work.

5. 6 genes and 16 miRNAs "which vary in competitive behavior specifically in presence of SARS-CoV-2 RNA" should be experimentally validated to show that the results are due to competition and not any other possible mechanism.

Validity of the findings

In the abstract, it is mentioned that “These transcripts have associations with COVID-19-related symptoms suggesting that our perturbation ability perspective has the potential to reveal the molecular basis of SARS-CoV-2 induced systemic dysfunctions.” This should be further explained in discussion. What type of molecular basis of SARS-CoV-2? Would they be helpful for development of new therapies or drugs? Impact and novelty of the study is not clearly stated.

Reviewer 3 ·

Basic reporting

This research uses Single-cell RNA-Seq and bulk miRNA-Seq data of Severe acute respiratory syndrome-related coronavirus(SARS-CoV-1) and Severe acute respiratory syndrome coronavirus (SARS-CoV-2) infected human cell lines to generate ceRNA network simulations using the ceRNAnetsim R package. miRNA:target interaction networks are constructed using miRTarBase (version 8) and Human miRNAs targeting SARS-CoV-1 and/or SARS-CoV-2 genome are extracted from a previous report (Khan et al., 2020). This study tried to examine the regulatory role of miRNAs and viral RNA in SARS-CoV-2 infections and identified 6 genes and 16 miRNAs that vary in competitive behaviour, specifically in presence of SARS-CoV-2 RNA.
The experimental conditions of the two study datasets are not explained in details. Why only two datasets were selected? How relevant are they to support the study hypothesis?
There should have been a flowchart to show the study pipeline to make general readers understand and follow the study easily.
How did Khan et al. arrived at the list of miRNAs potentially targeting the SARS genomes should be explained.
The paragraph starting at line 38 looks orphan and out of context and the statement is made without any reference too.
The introduction lacks review of related studies already reported in the literature.
The manuscript requires language corrections. for example, Figure 1 legend "p value < 0,05)" should be "p-value < 0.05", and many others.

Experimental design

The interaction of host miRNAs and viral RNAs has been predicted by several groups.
Sufficient attempt has not been made to connect the findings with previous reports.
It is not clear from the results how the findings fill an identified knowledge gap.
Functional annotation of miRNAs is not thorough and sounds incomplete and vague. For example, miRNAs have been reported to be associated with different disorders (liver, etc.), without giving specific details and pathways.
Figure 2- It is not clear why carcinoma cases were excluded during calculations. Please explain.

Validity of the findings

The study lacks any experimental validation and sounds rather too descriptive. It could have been rigourously linked to the already conducted computional and experimental studies, as a validation.
The impact and novelty of the study are not highlighted properly.
The study limitations are not mentioned or discussed.

---

## Round 0.2 · accepted · Accept

Thanks for making all the changes to the manuscript.